# Socio-economic impacts of the COVID-19 pandemic on new mothers and associations with psychosocial wellbeing: Findings from the UK COVID-19 New Mum online observational study (May 2020-June 2021)

**Emeline Rougeaux** *, **Sarah Dib**, **Adriana Vázquez-Vázquez**, **Mary S. Fewtrell**, **Jonathan C. K. Wells**

Childhood Nutrition Research Centre, Population, Policy & Practice Research and Teaching Programme, UCL Great Ormond Street Institute of Child Health, London, United Kingdom

* e.rougeaux@ucl.ac.uk

**Data Availability Statement:** We do not have consent from COVID-19 New Mum Study

## Abstract

Studies have reported unequal socio-economic impacts of the COVID-19 pandemic and associated restrictions in the UK, despite support packages. It is unclear how women with young children, a vulnerable group economically and psychosocially, havebeen impacted by income and employment pandemic changes, and how this is associated with psychosocial wellbeing. Using the UK COVID-19 New Mum online survey of women with children <12 months (28th May 2020-26th June 2021; N = 3430), which asked about pandemic impact on their i.ability to pay for rent, food, and essentials expenses separately, ii. employment (and/or partner's), and iii.past week mood, feelings and activities, we explored associations of i. & maternal age, household structure and income, i. & ii., and i. & iii. using logistic (odd ratios), multivariate (relative risk ratios/RRR), and linear (coefficients) regression respectively, and associated p-values. Overall, 30–40% of women reported any impact on ability to pay for expenses. Household earning <£20,000/yr had 6/4/7 times the odds of reporting an impact on food/rent/essentials (vs. > = £45,000/yr; p<0.001). Expenses impacts were associated with greater risk of partner business stopped/shut down (RRR:27.6/9.8/14.5 for rent/food/essentials [p<0.001 vs. no impact on employment]) or being made unemployed (RRR:15.2/9.5/13.5 [p<0.001]). A greater expenses impact was associated with higher (unhealthy) maternal psychosocial wellbeing score (coef:0.9/1.4/1.3 for moderate-major impact on rent/food/essentials vs. no impact [p<0.001]). The pandemic increased financial insecurity and associated poorer psychosocial wellbeing in new mothers. This is concerning given their pre-existing greater risk of poorer mental health and the implications for breastfeeding and child health and development. These findings reflect highlight the need for the UK government to assess shortfalls of implemented pandemic support policies and the provision of catch-up and better support for vulnerable groups such as new mothers, to avoid increasing socio-economic inequalities and the burden of poor maternal mental health and subsequent negative impacts on child wellbeing.

participants to make the raw data publicly available. However, justified requests for access to anonymized data can be sent to the UCL Ethics Committee (ethics@ucl.ac.uk), who will review these together with the study Principal Investigator Professor Jonathan Wells.

**Funding:** This study received no direct funding. All research at Great Ormond Street Hospital NHS Foundation Trust and UCL Great Ormond Street Institute of Child Health is made possible by the NIHR Great Ormond Street Hospital Biomedical Research Centre. The views expressed are those of the author(s) and not necessarily those of the NHS, the NIHR or the Department of Health.

**Competing interests:** The authors declare no conflict of interest with respect to the UK COVID-19 New Mum Study or this manuscript. MF receives an unrestricted donation for research on infant nutrition from Philips which is directly related to neither.

## Introduction

In late 2019, the emergence of the Severe Acute Respiratory Syndrome Coronavirus 2 (SARS-CoV-2) caused the global, ongoing pandemic of the Coronavirus 2019 (COVID-19) leading to over 6 million deaths worldwide as of early 2022, including over 170,000 deaths in the UK [1]. To limit the spread of the virus in the UK, various measures were implemented including phases of national lockdown and restrictions, workplace and school closures, and SARS-CoV-2 testing. Alongside these, broader social policies were also implemented to buffer the socio-economic impacts of the virus and the public health interventions aiming to combat its spread.

In the UK, 'Budget 2020' was unveiled in March 2020 which included statutory sick pay extensions for individuals needing to self-isolate, a hardship fund for local authorities to provide additional support to economically vulnerable households, business rates relief and interruption loan scheme, grant funding for small businesses in addition to financial support to the NHS and other public services [2]. The same month another package was introduced to protect jobs and incomes through increases in universal credit and tax credits and rent support, VAT payment deferrals and a job retention (furlough) scheme covering up to 80% of workers' wages (backdated to 1st of March 2020; then extended to September 2021). The Coronavirus Act 2020 was introduced on March 25th to help protect renters from eviction by extending the notice period from 2 to 3 months [2].

Despite these measures, research has highlighted the unequal impact of COVID-19 on different socio-economic groups [3]. Much of this variability was found, however, to be driven by existing disparities in health and the social determinants of health [3,4]. These findings emphasized the role of social determinants such as poor housing, malnutrition, poor air quality and discrimination in increasing the vulnerability to COVID-19 and its effects. For example, those of minority ethnic groups and deprived backgrounds may be more likely to suffer from chronic illnesses such as cardiovascular disease and asthma or to live in polluted environments, all of which have been found to increase the risk of severe COVID-19 [4].

Households with young children are likely to feel the socio-economic impacts of the pandemic the most. Data from a Social Metrics Commission Report from 2018/19 showed that families with children, particularly lone parent families and those with younger children, were more likely to be in poverty compared to other families [5]. A longitudinal study also found that during the pandemic, working parents with younger children fared worse in terms of financial wellbeing, and that mothers felt relatively harsher financial hardship than fathers [6]. Other studies have also shown that during the first weeks of the first UK lockdown there were clear inequalities in the impact of COVID-19, particularly relating to finances and basic needs like food, medication, and accommodation, with those in the lowest socio-economic position reporting the most adversities [7]. The economic repercussions of the pandemic will likely exacerbate inequalities further [8].

Studies have also shown important gender differences in the impact of the pandemic. Although COVID-19 mortality has been higher in men, evidence is increasingly suggesting that women were more likely to bear the brunt of the socio-economic impacts of the virus [9,10]. Our previous research highlighted a large burden of poor psychosocial wellbeing in new mothers during the pandemic which was exacerbated in those who had to travel to work, were more deprived or had seen a greater impact on their ability to buy food [11].

A study from the USA also found that giving birth during the pandemic was associated with higher levels of stress which in turn had a negative impact on maternal mental health, mother-child bonding and breastfeeding outcomes [12]. This adds to any potential stresses which can occur with the experience of pregnancy and birth and caring for a new child [13]. Not only can stress have a detrimental effect on maternal mental and physical health, but it

can also have a negative impact on child wellbeing through effects on mother-child bonding and relationships, breastfeeding outcomes and child physiological stress response [14–16].

This research aims to explore the impact of the pandemic on the socio-economic status of households with mothers caring for infants, and associations with maternal psychosocial wellbeing in the UK COVID-19 New Mum Study.

## Materials & methods

### 1. Data

The UK COVID-19 New Mum Study is a study of mothers with infants carried out during the COVID-19 pandemic from May to June 2020. It consisted of an online survey which was launched on May 27th, 2020, and primarily advertised via social media platforms such as Facebook, Twitter and Instagram and online groups used by mothers such as parental/maternal support groups and infant feeding groups. Mothers who were living in the UK, aged 18 years or older and who had an infant under 12 months of age were eligible to participate. The survey was designed using RedCap online software.

### 2. Ethics statement

Ethical approval was obtained from the UCL Research Ethics committee (0326/017). The landing page of the survey provided participants with information about the study, ethical processes and contact details which participants were asked to read before giving consent and proceeding with the survey. The survey was entirely anonymous; participants could choose not to respond to any questions they felt uncomfortable answering. At the end of the survey, participants were also provided with a list of resources for mental health and infant feeding support, including some tailored to specific groups with protected characteristics.

### 3. Measures

The survey explored demographic and socio-economic background characteristics of the mother and her household, the perceived impacts of COVID-19 and infant birth and feeding practices which are described in more detail elsewhere [17]. For the present study, self-reported data from the full survey (27th May 2020-26th June 2021) was used, and the variables described in Table 1 were analysed.

A measure of maternal psychosocial wellbeing was created using Principal Component Analysis (PCA) following the same methodology described in a previous publication [11]. This was carried out on 12 variables reflecting maternal mood, feelings, time allocation and opportunities in the week preceding the survey which were introduced into the survey from August 1st. These were obtained by asking the mother "In the last week, how much do the following statements apply to you?" followed by 12 negative (e.g I've been feeling down) and positive statements (e.g I feel able to cope with the situation) and responses were categorised as not at all, very little, to some extent, or to a high extent. These are shown in S1 Table. An overall Kaiser-Meyer-Olkin measure of sampling adequacy of 0.9 suggested the data was suitable for PCA.

A higher score indicates poorer psychosocial wellbeing overall, that is a *greater likelihood* of low mood, negative feelings and behaviours; and a *lower likelihood* of coping and having had the opportunity to socialize and take part in positive activities.

### 4. Analyses

**a. Sample characteristics.**   Using the self-reported survey data, we looked at characteristics of the sample of respondent mothers and their households (given as percentages and

**Table 1. Study measures.**

| Measure | Description |
|---|---|
| Background demographic and socio-economic characteristics: | |
| Maternal age | In years. |
| Child age | In years. |
| Maternal self-reported ethnicity | Categorized as White/Caucasian/European, Mixed, Asian, Black British/African/Caribbean/Other, Arab, Latino or Other. |
| Maternal highest academic degree obtained | Categorized as Less than 5 GCSEs A-C grade, 5 or more A-C grade GCSEs, A-levels/High school diploma, Bachelor's degree, Master's degree, and Doctoral or professional degree. |
| Household structure | Categorized as married/civil partnership/cohabitating, lone parent living on their own, and lone parent living with family. |
| Number of children (under 18 years) in the household | Categorized as 1, 2–3, and 4 or more. |
| Yearly joint family income including benefits and before taxes | Categorized as <£20,000, ≥£20,000 & <£30,000, ≥£30,000 & <£45,000, ≥£45,000 & <£75,000, ≥£75,000 & <£100,000, and >£100,000. |
| Socio-economic impacts of COVID-19: | |
| Impact on household ability to cover food, rent/mortgage and essential (such as utilities or medication) expenses | Response options for each were: no impact, minor impact, moderate impact, major impact or too soon to tell. In this study responses were either classed into three categories (no impact, minor impact or moderate/major impact) or two categories (no impact, any impact). |
| Type of impact on employment (mother's, and partner's if applicable) | Responses for both were categorised as: no change, now working remotely, own business closed or shut down, made redundant, put on furlough, affected in other ways. |
| Other impact on employment | Mothers were given the choice to give further information on how COVID-19 had impacted their and/or their partner's employment as free text. |
| Wellbeing outcomes: | |
| Maternal psychosocial wellbeing | A measure created using Principal Component Analysis (PCA) on 12 variables reflecting maternal mood, feelings, time allocation and opportunities in the week preceding the survey which were introduced into the survey from August 1st. Methods are described in more detail in the text. A higher score indicates poorer psychosocial wellbeing. |

numbers) and, to assess representativeness, we compared these with the latest available national data on women of childbearing age, which was either from either the latest Census (2011; England & Wales only) or Households Above Average Income Survey (2019/20; UK).

**b. Impact of the pandemic on ability to cover household expenses.** We first explored associations of demographic and economic factors (child age, maternal ethnic group, education, number of children, region, maternal age, household structure and income) with the self-reported measures on the impact of the pandemic on the ability of the woman's household to cover food, rent/mortgage, and essentials expenses using Pearson Chi2 tests for association (and 95% Confidence Intervals [CIs]).

For demographic and economic factors found to be associated (maternal age, household structure, income), we further explored these in relation to impact on food, rent/mortgage, and essentials expenses using multiple logistic regression (results given as odds ratios [OR] and their 95% CIs).

We also explored associations of impact on expenses (separately for food, rent/mortgage, and essentials) by survey period (May 2020, June-July 2020, August 2020 or Sept 2020-June

2021), adjusting for significant demographic and economic factors (maternal age, household structure, income) using multiple logistic regression (results given as odds ratios [OR] and their 95% CIs).

**c. Impact of the pandemic on employment.** We first assessed associations of demographic and economic factors with self-report impact of COVID-19 on mother's and partner's (if applicable) employment using Pearson Chi2 tests for association (results given as p-values and 95% CIs).We then assessed associations of the reported impact on employment and the reported impact on household expenses (for each food, rent/mortgage, and essentials separately) using multinomial regression (results given as relative risk rations [RRRs] and their 95% CIs, adjusted for confounders [maternal age, as partner's age was not available, and household income]).

Finally, we collated free text responses reporting other impacts of COVID-19 on employment-related factors and summarized the main themes from these.

**d. Associations of pandemic expenses impact and maternal psychosocial wellbeing.** We assessed associations of reported impacts of the pandemic on household expenses (for each food, rent/mortgage, and essentials separately) and maternal psychological wellbeing scores first using ANOVA tests of association (given as p-values) and then linear regression adjusted for confounders (maternal age, household structure, income; the results of which are given as coefficients and their 95% CIs).

All analyses were carried out in Stata/SE 15.1 (StataCorp. 2017. Stata Statistical Software: Release 15. College Station, TX: StataCorp LLC).

## Results

### 1. Sample characteristics

A total of 3430 women resident in the UK with infants aged 12 months or less responded to the survey between 28[th] of May and 16[th] of June 2021. The sample was diverse with women from all or most age groups, regions, ethnicities, education, and income levels (Table 2). A majority of women (88%) were aged 26–40 years old, were of white ethnicity (91%), had an infant below 6 months of age (60%), had achieved a minimum of A-levels or high school diploma (87%), were partnered (married, in a civil partnership or cohabiting; 94%), only had one child (including the survey child; 64%), were in households with a yearly income higher or equal to £45,000 (60%) and were living in England (90%, of which 16% in Greater London; Table 2). Compared to the latest available national data on women of childbearing age, the UK Covid-19 New Mum Study sample contained fewer mothers in the younger and oldest age groups, more mothers of White/Caucasian/European ethnicity, with higher education, in married/civil partnership/cohabiting relationships, and who were first time mothers. There were also small differences by postcode region of residence; mothers in our study were more likely to live in South West England & Channel Islands, the West Midlands, Greater London, and South-East England but less likely to live in other areas of England, Scotland, Northern Ireland and Wales (Table 2).

### 2. Impact on ability to cover household expenses

Overall, we find that 34% of the mothers in the survey reported that the COVID-19 pandemic had an impact (20% minor, 14% moderate/major) on their household's ability to pay for *food*, while 39% reported an impact (19% minor, 20% moderate/major) on the ability to make *rent or mortgage payments*, and 31% reported an impact (17% minor, 13% moderate/major) on the ability to pay for *essentials* such as medicine or utilities (Fig 1 and S2 Table).

**Table 2. Characteristics of mothers and infants in the UK COVID-19 New Mum Study (28ᵗʰ May 2020-26ᵗʰ June 2021); % (N = 3430).**

| | | UK Covid-19 New Mum Study | National data ([1] Census 2011 England & Wales; [2] Households Below Average Income 2019/2020 UK) |
|---|---|---|---|
| Maternal age (years) | 18–25 | 10% (344) | 18% (18–24 years) [1] |
| | 26–30 | 29% (1005) | 14% (25–29 years) [1] |
| | 31–35 | 38% (1298) | 13% (30–34 years) [1] |
| | 36–40 | 20% (670) | 13% (35–39 years) [1] |
| | 41–52 | 3% (112) | 42% (40–49 years) [1] |
| | *Missing (n)* | *1* | - |
| Maternal ethnicity | White/Caucasian/European | 90% (3023) | 84% [1] |
| | Mixed | 3% (110) | 2% [1] |
| | Asian | 4% (139) | 9% [1] |
| | Black British/African/Caribbean/Other | 2% (55) | 3% [1] |
| | Arab, Latino & Other | 1% (27) | 1% [1] |
| | *Missing (n)* | *76* | |
| Infant age (months) | <2 | 13% (439) | - |
| | 2 to <3 | 24% (813) | - |
| | 4 to <6 | 24% (821) | - |
| | 6 to <8 | 16% (557) | - |
| | 8 to <10 | 13% (440) | - |
| | 10–12 | 11% (360) | - |
| | *Missing (n)* | *0* | - |
| Maternal highest degree obtained | Less than 5 GCSEs A-C grade | 5% (168) | 22% [1] |
| | 5 or more A-C grade GCSEs | 8% (274) | 25% [1] |
| | A-levels/High school diploma | 22% (734) | 18% [1] |
| | Bachelor's degree | 40% (1340) | 35% (Bachelor's degree or higher) [1] |
| | Master's degree | 15% (491) | - |
| | Doctoral or professional degree | 10% (334) | - |
| | *Missing (n)* | *89* | - |
| Household structure | Married/civil partnership/cohabitating parents | 91% (3232) | 78% [2] |
| | Lone parent, living on own | 4% (123) | 22% [2] |
| | Lone parent, living with family | 2% (77) | - |
| | *Missing (n)* | *98* | |
| Total number of children <18 years in the household (including all ≤12 months) | One | 64% (1926) | 25% [2] |
| | Two-three | 32% (965) | 46% (two) [2] |
| | Four or more | 4% (131) | 29% (three or more) [2] |
| | *Missing (n)* | *408* | |
| Household income (yearly) | < £20,000 | 8% (250) | 32% [2] |
| | ≥£20,000 and <£30,000 | 11% (348) | 41% [2] |
| | ≥£30,000 and <£45,000 | 21% (641) | 21% [2] |
| | ≥£45,000 and <£75,000 | 34% (1069) | 7% (£45,000 and over) [2] |
| | ≥£75,000 and <£100,000 | 14% (436) | - |
| | >£100,000 | 12% (381) | - |
| | *Missing (n)* | *305* | - |

*(Continued)*

 

**Table 2.** (Continued)

| | | UK Covid-19 New Mum Study | National data ([1] Census 2011 England & Wales; [2] Households Below Average Income 2019/2020 UK) |
|---|---|---|---|
| Postcode district of residence | South West England & Channel Islands | 12% (406) | 8% [2] |
| | East England | 10% (326) | 10% [2] |
| | East Midlands | 7% (230) | 7% [2] |
| | West Midlands | 13% (430) | 9% [2] |
| | Greater London | 16% (521) | 14% [2] |
| | North West England & Isle of Man | 9% (297) | 11% [2] |
| | North East England | 8% (254) | 12% [2] |
| | South East England | 15% (499) | 14% [2] |
| | North Ireland | 1% (47) | 3% [2] |
| | Scotland | 5% (180) | 7% [2] |
| | Wales | 4% (121) | 5% [2] |
| | *Missing (n)* | 119 | |
| Survey response period | May 2020 | 27% (927) | - |
| | June-July 2020 | 32% (1100) | - |
| | August 2020 | 36% (1221) | - |
| | September-June 2021 | 5% (182) | - |
| | *Missing (n)* | 0 | - |

We find a graded and significant association (Chi2 test p-values <0.001) between maternal age and the reported impact of COVID-19 on household ability to cover all types of expenses (food, rent/mortgage and essentials), with younger mothers (aged 18–25 years) being more like to report an impact (both minor and moderate/major) across all three categories. For example prevalences of a moderate-high impact for mothers aged 18–25 years and 36+ years were 24% versus 12% for food expenses, 28% versus 18% for rent/mortgage, and 21% versus 12% for essentials (Fig 1 and S2 Table). We also see that lone mothers had a higher prevalence across all types of expenses of reporting both a minor and moderate/major impact of the pandemic on household expenses compared to partnered mothers. Twenty seven percent of lone mothers and 13% of partnered mothers reported a moderate-major impact on food expenses; for rent/mortgage this was 20% and 25%, and for essentials 13% and 21% (Chi2 test p-value < 0.001; S2 Table).

Households with an income < £45,000 reported a greater impact (both low or moderate-major) on expenses compared to households with an income ≥ £45,000. We also find that households with an income < £20,000 had the highest prevalence of reporting a moderate/major impact of the pandemic on ability to pay for rent/mortage (at 38%) and essentials (at 35%) (Chi2 test p-value < 0.001; S2 Table) but they were similar to other households below < £45,000 for food expenses.

Child age, maternal ethnic group, education, number of children and region were not found to be associated with impact on ability to cover expenses and are not shown here.

However, when all three measures (maternal age, household structure and income) are simulateously explored in regression analyses (with impact as a binary variable), only income remains an important associated factor with impact of the pandemic on household expenses (Table 3).

Overall we find a graded association of income with impact on expenses, with a higher odds of reporting any impact the lower the household income (Table 3). Notably, after

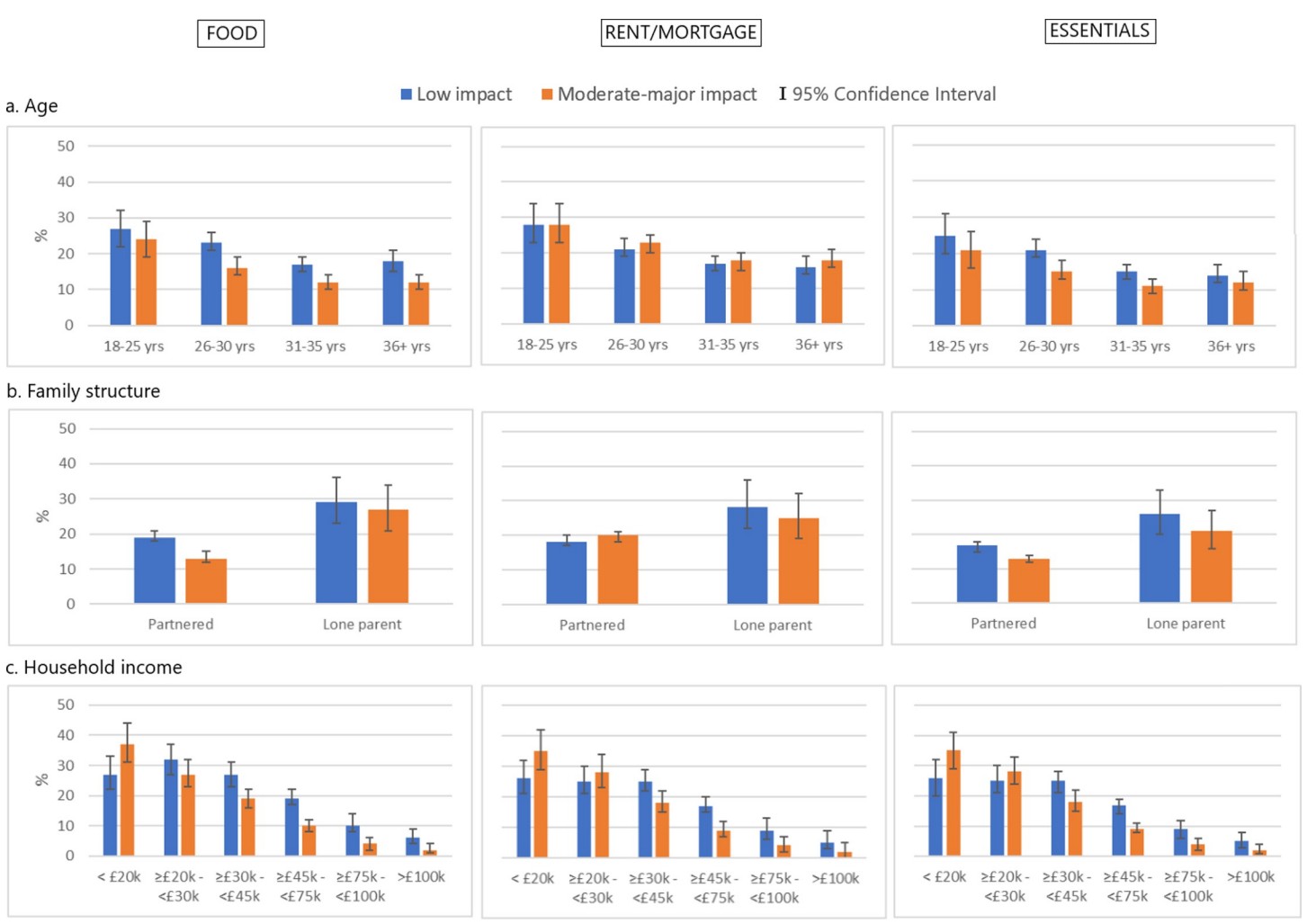

**Fig 1. Impact of the COVID-19 pandemic on household ability to cover food, rent/mortgage, and essentials expenses by maternal socio-demographic characteristics in the COVID-19 New Mum Study; %.**

controlling for maternal age and family structure, households with an income < £20,000 have roughly 6 times the odds of reporting an impact on food and essentials expenses and 4 times the odds of reporting an impact on rent/mortgage compared to households earning ≥ £45,000 (Table 3).

The odds of reporting an impact on expenses appears to increase over time, with no change after adjusting for income, family structure and maternal age. Respondents in August 2020 had 1.3, 1.4 and 1.5 times the odds of reporting an impact on rent, food and essentials expenses respectively compared to respondents in May 2020 (p < 0.05 for rent and food, p < 0.001 for essentials, Table 4). Although the odds ratio also appeared similarly higher in September 2020 (compared to May 2020), this was only significant for food expenses. There were no apparent differences in June-July 2020.

## 3. Changes in employment

**Maternal employment.** Of the mothers who participated in the study, just under 90% did not report a change in employment during the pandemic due to being on maternity leave

**Table 3. Multiple regression of the COVID-19 pandemic impact on ability to pay for expenses by maternal age, household structure and income in the UK COVID-19 New Mum Study (28th May 2020-26th June 2021).**

| | | Unadjusted | | Adjusted | |
|---|---|---|---|---|---|
| | | OR | 95% CI | OR | 95% CI |
| **Impact on buying food (N = 2989)** | | | | | |
| Maternal age | 18–25 years | 2.4** | 1.8–3.2 | 1.2 | 0.9–1.7 |
| | 26–30 years | 1.5** | 1.2–1.9 | 1.2 | 0.9–1.5 |
| | 31–35 years | 0.9 | 0.8–1.2 | 0.9 | 0.7–1.2 |
| | 36 year and over | Reference group | | | |
| Household structure | Lone mother | 2.6** | 2.0–3.6 | 1.0 | 0.7–1.4 |
| | Partnered mother | Reference group | | | |
| Household income (yearly) | < £20,000 | 6.7** | 5.0–8.9 | 6.2** | 4.4–8.6 |
| | ≥£20,000 and <£45,000 | 3.8** | 3.2–4.5 | 3.6** | 3.0–4.3 |
| | ≥£45,000 | Reference group | | | |
| **Impact on rent/mortgage payments (N = 3096)** | | | | | |
| Maternal age | 18–25 years | 2.5** | 1.9–3.2 | 1.4 | 1.0–1.9 |
| | 26–30 years | 1.5** | 1.2–1.8 | 1.1 | 0.9–1.4 |
| | 31–35 years | 1.0 | 0.8–1.2 | 0.9 | 0.7–1.1 |
| | 36 year and over | Reference group | | | |
| Household structure | Partnered mother | 1.9** | 1.4–2.5 | 0.8 | 0.5–1.1 |
| | Lone mother | Reference group | | | |
| Household income (yearly) | < £20,000 | 4.1** | 3.1–5.5 | 4.1** | 2.9–5.7 |
| | ≥£20,000 and <£30,000 | 2.8** | 2.4–3.3 | 2.79** | 2.2–3.2 |
| | ≥£45,000 | Reference group | | | |
| **Impact on essentials expenses (N = 3085)** | | | | | |
| Maternal age | 18–25 years | 2.4** | 1.8–3.2 | 1.2 | 0.8–1.6 |
| | 26–30 years | 1.6** | 1.3–2.0 | 1.2 | 0.9–1.5 |
| | 31–35 years | 1.0 | 0.8–1.2 | 0.9 | 0.7–1.2 |
| | 36 year and over | Reference group | | | |
| Household structure | Partnered mother | 2.1** | 1.5–2.8 | 0.8 | 0.5–1.1 |
| | Lone mother | Reference group | | | |
| Household income (yearly) | < £20,000 | 6.6** | 4.9–8.8 | 6.8** | 4.8–9.5 |
| | ≥£20,000 and <£30,000 | 3.8** | 3.2–4.5 | 3.7** | 3.0–4.4 |
| | ≥£45,000 | Reference group | | | |

Table footnotes: ** p < 0.001,

* p < 0.05.

(72%), being unemployed prior (8%) or for other unspecified reasons. Of mothers who were employed when the pandemic occurred (including those on maternity leave), 2% changed to remote working, 2% were put on furlough, 1% were made unemployed, 1% saw their own business closed/shut down and 4% were impacted in other ways.

Numbers were too small to fully explore changes in employment across socio-economic characteristics; however, prevalence comparisons between mothers in households with an income < £45,000 and ≥ £45,000 suggest those in lower incomes were less likely to change to work from home (1.3% vs 3.2%) but more likely to be put on furlough (3.3% vs 1.4%), be made unemployed (1.8% vs 0.9%) or have their business stopped/shut down (1.2% vs 0.8%) (Pearson Chi2 test p-value < 0.001). Lack of power also meant regression analyses could not be carried out to explore associations with impact on the ability to pay for household expenses.

**Table 4. Multiple regression of impact of the COVID-19 pandemic on expenses over different periods of the UK COVID-19 New Mum Study (28th May 2020-26th June 2021).**

| | | Unadjusted | | Adjusted for income, family structure and maternal age | |
|---|---|---|---|---|---|
| | Time period | OR | 95% CI | OR | 95% CI |
| **Impact on buying food** (any impact vs. none; N = 2989) | May 2020 | *Reference group* | | | |
| | June-July 2020 | 0.9 | 0.8–1.1 | 1.0 | 0.8–1.2 |
| | August 2020 | 1.4* | 1.1–1.7 | 1.4* | 1.1–1.7 |
| | Sept 2020-June 2021 | 1.6* | 1.1–2.4 | 1.7* | 1.1–2.5 |
| **Impact on rent/mortgage payments** (any impact vs. none; N = 3096) | May 2020 | *Reference group* | | | |
| | June-July 2020 | 0.9 | 0.7–1.1 | 0.9 | 0.7–1.1 |
| | August 2020 | 1.3* | 1.1–1.6 | 1.3* | 1.1–1.6 |
| | Sept 2020-June 2021 | 1.5 | 1.0–2.2 | 1.5 | 1.0–1.6 |
| **Impact on essentials expenses** (any impact vs. none; N = 3085) | May 2020 | *Reference group* | | | |
| | June-July 2020 | 0.9 | 0.7–1.1 | 0.9 | 0.8–1.2 |
| | August 2020 | 1.5** | 1.2–1.8 | 1.5** | 1.2–1.8 |
| | Sept 2020-June 2021 | 1.5 | 1.0–2.2 | 1.5 | 1.0–2.3 |

Table footnotes: ** p < 0.001,

* p < 0.05.

**Partner employment.** Of mothers with partners who were employed when the pandemic started (98% of all partners), 58% reported their partner's employment had not been affected by the pandemic while 14% reported they had changed to working remotely, 11% that they were put on furlough, 2% that they were made unemployed, 4% that they'd had their business closed or shut down and 11% that their employment was impacted in other ways. This was associated with household income; those in lower income groups appeared to have lower percentages who changed to remote working but higher percentages who were made unemployed, put on furlough, had their business closed/shut down and had their employment impacted in other ways (Fig 2 and S3 Table).

The regression analyses of change in partner's employment and impact on ability to cover household expenses showed that both were strongly associated, and this changed little when adjusting for possible confounders (maternal age [assumed to reflect partner age which was not available] and household income). All changes in partner employment were associated with an increased risk of reporting an impact of the pandemic on ability to cover household expenses compared to those who reported no change in partner employment. The greatest risk increases were seen for households where the partner was made unemployed or had their business closed/shut down, and particularly for rent/mortgage and essentials expenses (Table 5).

**Qualitative reports of employment-related impact.** Several mothers reported other changes to employment and related factors for themselves and/or partner as free text. These responses are summarised in S4 Table. For most part mothers reported negative impacts on the household and stress and anxiety related to this, reflecting previous findings that a large proportion of mothers in the study reported low mood and anxiety during the lockdown [11,17]. Struggles or changes relating to balancing work and childcare due to closures and fears of being made unemployed (mostly after the end of maternity leave) were the most frequent issues reported by mothers. For example, some mothers highlighted not being able to work, needing to extend maternity leave (without pay), or needing to take unpaid leave due to limited childcare options during the pandemic. On another hand, others reported needing to

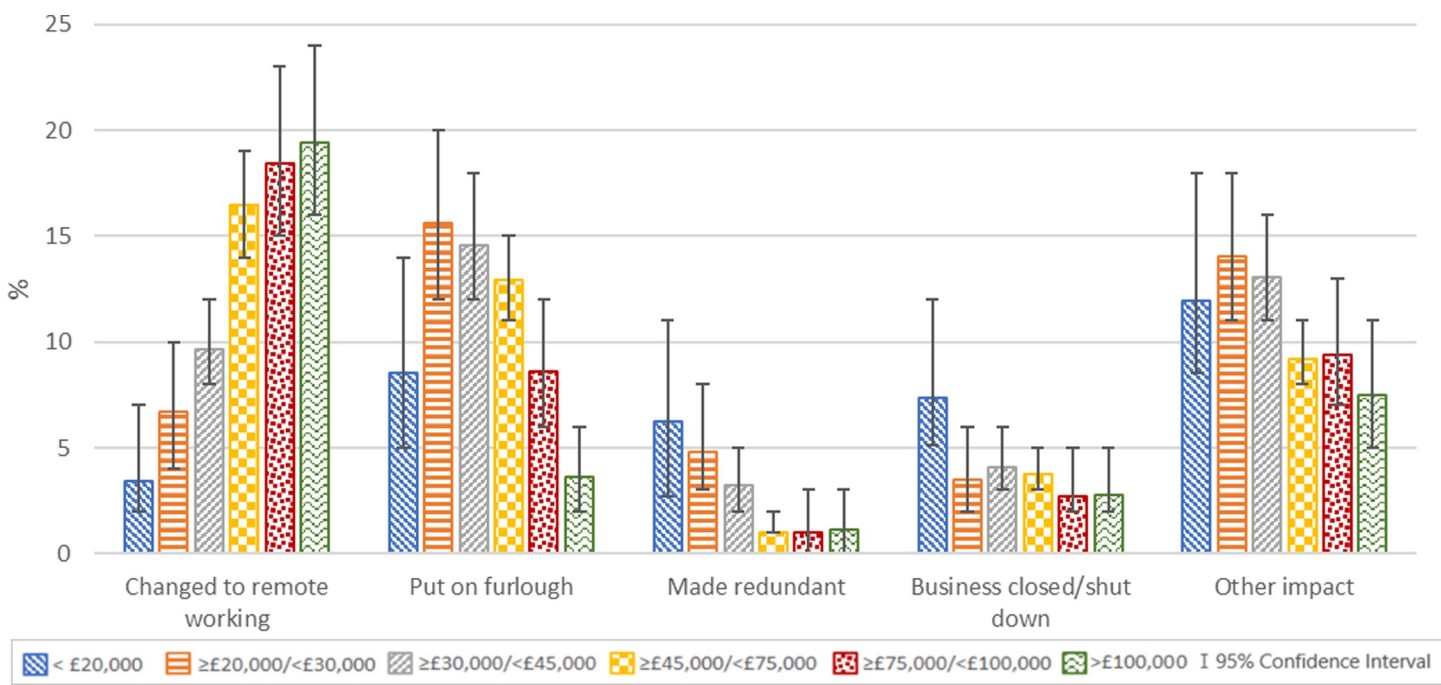

**Fig 2. Impact of the COVID-19 pandemic on partner employment by household income in the COVID-19 New Mum Study; %.**

return to work earlier than planned to make up for partner's income loss. For their partners, the most reported was a reduction in work hours or opportunities (e.g if free-lance or self-employed), salary cuts and being put on unpaid leave. A few mothers highlighted the lack of

**Table 5. Regression of partner's change in employment and impact on expenses during the COVID-19 pandemic, UK COVID-19 New Mum Study (28th May 2020-26th June 2021, N = 3232).**

| | | Food expenses (impact vs. none) | | Food expenses (impact vs. none) | | Essentials (impact vs. none) | |
|---|---|---|---|---|---|---|---|
| | | RRR | 95% CI | RRR | 95% CI | RRR | 95% CI |
| **Unadjusted** | | | | | | | |
| Change in partner's employment | None | Reference group | | | | | |
| | Changed to remote working | 1.2 | 0.9–1.5 | 1.2 | 0.9–1.5 | 1.1 | 0.9–1.5 |
| | Put on furlough | 3.5** | 2.7–4.4 | 5.4** | 4.2–7.0 | 4.3** | 3.4–5.5 |
| | Made unemployed | 10.4** | 5.8–18.8 | ** | 8.3–34.4 | 14.6** | 8.0–26.8 |
| | Business stopped or closed down | 7.1** | 4.7–10.8 | 24.0** | 12.7–45.2 | 10.3** | 6.7–16.0 |
| | Other change in employment | 4.0** | 3.1–5.1 | 4.2** | 3.2–5.5 | 3.8** | 3.0–5.0 |
| **Adjusted for maternal age and household income** | | | | | | | |
| Change in partner's employment | None | Reference group | | | | | |
| | Changed to remote working | 1.8** | 1.3–2.3 | 1.5* | 1.1–1.9 | 1.7* | 1.2–2.2 |
| | Put on furlough | 3.5** | 2.7–4.6 | 5.1** | 3.9–6.7 | 4.3** | 3.3–5.6 |
| | Made unemployed | 9.5** | 4.9–18.5 | 15.2** | 7.0–32.9 | 13.5** | 6.9–26.6 |
| | Business stopped or closed down | 9.8** | 6.0–15.9 | 27.6** | 14.0–54.3 | 14.5** | 8.7–24.3 |
| | Other change in employment | 4.2** | 3.2–5.6 | 4.0** | 3.0–5.4 | 4.1** | 3.0–5.4 |

Table footnotes: ** p < 0.001,

* p < 0.05.

support for parents who were unable to work due to having vulnerable children and needing to shield. Several mothers also reported their partners had longer hours; these were mostly in medical professions. Although most of the reported impacts were relating to financial strain, there were also some reports of lower debts and increased disposable income (S4 Table).

## 4. Impact on household expenses and maternal psychosocial wellbeing

Depending on the specific question, 1160 to1170 mothers responded to the twelve recent psychosocial wellbeing questions introduced in the survey in August 2020 (S1 Table); when combining these using PCA, a measure of psychosocial wellbeing was obtained for 1134 mothers. This measure is given as a score, with a higher number indicating poorer psychosocial wellbeing in the week preceding the survey.

We find the psychosocial score to be positively associated with all three impact on expenses measures (ANOVA p-values all < 0.001); that is women reporting a greater impact of COVID-19 on food, rent/mortgage and essentials expenses were more likely to also report poorer psychosocial wellbeing in the week before they completed the survey (Fig 3).

The regression results reflected these findings with a greater impact on expenses being associated with a significantly higher psychosocial wellbeing score, particularly with food and essentials expenses, which changed little after adjusting for confounders (maternal age, household income and family structure; Table 6).

Mothers who did not respond to the psychosocial wellbeing questions were more likely to be in the lowest income and younger age groups than mothers who did respond (not shown here).

## Discussion

Approximately a third of mothers in the study reported that their household's ability to pay for expenses had been impacted by the COVID-19 pandemic, of which just under half reported a moderate-major impact. A greater impact was reported for ability to pay rent/mortgage payments followed by food expenses and essentials expenses. Across all three types of expenses there was a graded association with household income. Notably, households in the lowest income category (< £20,000 per year) had 4 to 6 times the odds of reporting an impact (of any level) of the pandemic on their food, rent/mortgage, and essentials expenses to households

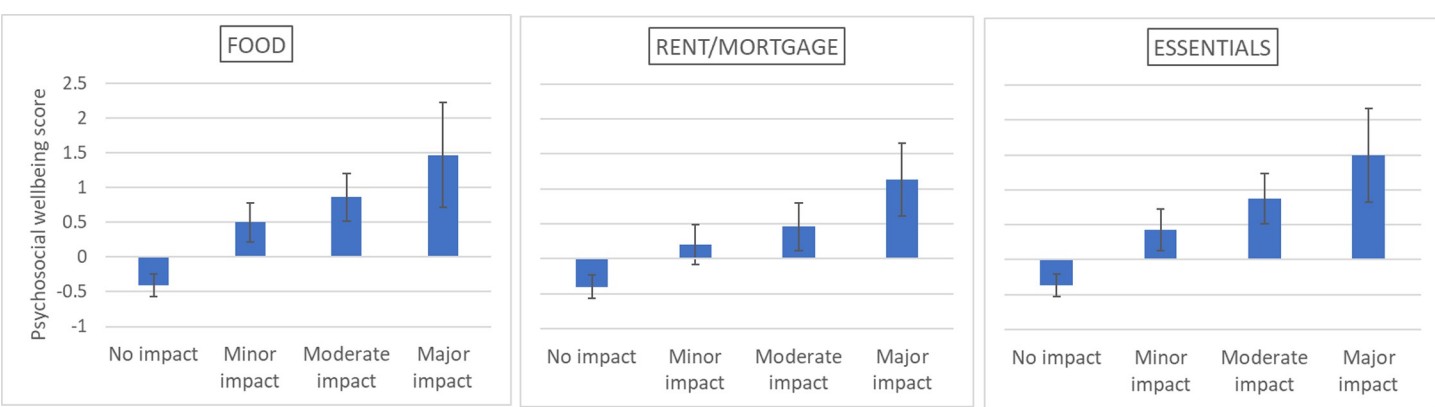

* A higher score indicates poorer wellbeing; obtained from Principal Component Analysis of 12 variables reflecting poor maternal mood, feelings, time allocation and opportunities in the week preceding the survey; $\mathrm{I}$ 95% Confidence Interval

**Fig 3. Maternal recent psychosocial wellbeing score by reported COVID-19 pandemic impact on household expenses in the COVID-19 New Mum Study.**

**Table 6. Regression of COVID-19 pandemic impact on ability to pay for household expenses by maternal psychosocial wellbeing[a] in the UK COVID-19 New Mum Study (28th May 2020-26th June 2021; N = 1134).**

| | | Unadjusted | | Adjusted (for maternal age, household income and family structure) | |
|---|---|---|---|---|---|
| | | Coefficient | 95% CI | Coefficient | 95% CI |
| **Impact on food expenses** | None | Reference group | | | |
| | Minor | 0.9** | 0.6; 1.2 | 0.9** | 0.5; 1.3 |
| | Moderate/major | 1.4** | 1.1; 1.8 | ** | 0.9; 1.8 |
| **Impact on rent/mortgage payment** | None | Reference group | | | |
| | Minor | 0.6* | 0.3; 1.0 | 0.5* | 0.2; 0.9 |
| | Moderate/major | 1.1** | 0.8; 1.4 | 0.9** | 0.5; 1.3 |
| **Impact on essentials expenses** | None | Reference group | | | |
| | Minor | 0.8** | 0.5; 1.1 | 0.7** | 0.3; 1.1 |
| | Moderate/major | 1.4** | 1.1; 1.8 | 1.3** | 0.9; 1.7 |

Table footnotes: ** $p < 0.001$,

* $p < 0.05$,

[a] A higher score indicates poorer wellbeing; obtained from Principal Component Analysis of 12 variables reflecting poor maternal mood, feelings, time allocation and opportunities in the week preceding the survey.

with a yearly income of £45,000 and over after controlling for maternal age and family structure. Lone and younger mothers were more likely to report an impact on expenses due to a higher likelihood of being in a lower income household. Women also had higher odds of reporting an impact on expenses, particularly for food, in later periods of the survey (August 2020 onwards compared to May 2020).

Although government projections suggest that household net income losses during the study period should have been adequately met by government income, job retention and welfare support, particularly for those in the lowest income groups, others have found that poorer families were disproportionately impacted socio-economically by the pandemic, due to pre-existing budget cuts but also to changes in essential services during this period such as child care and school closures and inconsistent free school meals provision [3,17,18]. A longitudinal study from the first weeks of the initial UK pandemic lockdown showed that those in lower socio-economic positions reported a greater number of adversities after adjusting for confounders [7]. In addition, Wright et al. found a socio-economic gradient in having seen a major cut in household income, being unable to pay bills, access sufficient food and required medication [7]. Our findings highlighting increasing difficulties in meeting household expenses over time and fears of future redundancies and increases in debt also suggest, as others have, that the long-term economic effects of the pandemic are likely to exacerbate pre-existing high precarity in families with children, thus pushing them further into poverty [3,19].

Furthermore, in our study we find that mothers who reported that their households were impacted by the pandemic in their ability to cover expenses, particularly those who reported a moderate/major impact on food and essentials expenses, were more likely to have poorer psychosocial wellbeing. In the Understanding Society longitudinal study, Cheng et al. found that with the emergence of the pandemic, greater financial insecurity was more likely to be reported in working adults with children, compared to without, and particularly if they were of lower income before the pandemic, had younger children, or were mothers. This was associated in turn with poorer mental health, particularly in lower income households [6]. These findings are not surprising given that being at risk of, and especially experiencing, financial hardship was found to be associated with a higher risk of developing mental health problems in time [20].

For mothers with young infants, these findings are of particular concern as, for some, the psychosocial impact of the pandemic may add to existing psychosocial stress related to pregnancy, birth and caring for a young child, particularly in light of COVID-19 measures which reduced crucial emotional and practical support by limiting partners for appointments and delivery, availability of postnatal services and social interactions [11,12,21,22]. Maternal stress in pregnancy may negatively impact child birthweight and, postnatally, it may affect the ability to bond and breastfeed effectively, which may in turn have long-term implications for development and health [14,23,24].

A report using data from March to May 2020 also suggested that greater levels of pre-pandemic poverty were associated with greater negative impacts on employment (such as reduced hours, earnings and/or being put on furlough or losing one's job) [25]. The longitudinal study by Wright et al. additionally showed a socio-economic gradient in participants (and/or their partner) having lost their job or been unable to work during the start of the pandemic [7]. In our study, as most mothers were on maternity leave when the pandemic started, few reported having their employment impacted compared to partners. However, prevalences suggested that both mothers and partners in lower income households were more likely to be made unemployed or have their own business stopped or closed. Of mothers with partners employed at the start of the pandemic, roughly 11% were put on furlough which is somewhat lower than official figures (15–30% depending on the region and work sector), likely due to issues of representation in our sample which we discuss further on [26]. In the qualitative sections of the survey, many mothers expressed fears of future redundancies, mentioned cuts to pay and work hours, and difficulties finding employment so it is possible that the inequalities we have found will increase over time. We also find that women whose partner had lost employment or seen their business closed/shut down were more likely to report not being able to meet household expenses compared to those who had been put on furlough or switched to working from home. Blundell et al have suggested that those made unemployed due to the pandemic would likely have been less supported than those furloughed due to being reliant on pre-existing benefit systems rather than pandemic support packages, thus potentially increasing inequalities between these groups [3].

Several mothers reported additional issues relating to childcare, including not being able to work from home with a young child, needing to extend maternity leave or take unpaid leave to care for their child/children. A qualitative study of UK mothers carried out in summer 2020 highlighted similar difficulties in balancing home-schooling and work as well as resulting feelings of stress, guilt, and worry [27]. Another survey of almost 20 thousand mothers carried out in July 2020 suggested that half of respondents who had been made unemployed during the pandemic blamed lack of childcare provision [27]. Our findings also reflect those from an Institute of Fiscal Studies report from the first period of lockdown which found mothers were more likely to have lost or stopped work or seen their working hours reduced compared to fathers and, among those working from home, mothers were likely to simultaneously care for children than fathers during the pandemic [9,28]. We found that partners were more likely to have lost their work than mothers, but this is most likely because our study focussed on mothers with infants, most of whom were on maternity leave. Maternal concerns of future potential redundancies may indicate their unemployment could increase in the future.

Overall, our findings support those of petitions made to the UK government which highlight the continued lack of provision in pandemic support packages for new parents, which will likely have important consequences on the long-term wellbeing of children [29]. However, recommendations for increases in maternity leave duration, extensions of maternity exemption certificates (allowing free access to dental care and other benefits which could not be used with pandemic closures), access to parental health and support services were rejected on the

basis that existing support was sufficient [29]. The parties involved have recently renewed the call for greater government support, emphasizing the need for new parents to be able to catch-up on services they had missed out on in the pandemic, for planned mental health interventions and funding to be more targeted at new mothers, for greater protection from workplace discrimination and redundancy for new or soon-to-be parents, particularly women, and for more affordable access to childcare [29].

It is possible that we may have underestimated inequalities in the socio-economic impacts of tge COVID-19 pandemic due to lack of representation of different parts of the UK and different groups of the population. For example, our sample was primarily located in England, more likely to be of white ethnicity, wealthier and more educated than the general population (Table 2). Recent research has shown that not only did Black, Asian and minority ethnic migrant in the UK have higher levels of poverty in pre-pandemic times, but they also experienced greater financial hardship during the pandemic [30]. Small group sample sizes however meant we could not compare different ethnic groups. We were also limited in our ability to explore change over time as very few women responded to the survey from September 2020 onwards. Access to our survey may have been limited for some women due to it being self-completed, only available in English and online, thus further affecting how representative our sample might be, and therefore potentially our results.

We may also have underestimated the association of impact of the pandemic on ability to cover expenses and maternal psychosocial wellbeing as a number of mothers did not answer the questions used to assess psychosocial wellbeing and these mothers were found to be younger and in lower income households, both characteristics which our current findings and previous research suggest may be associated with more negative socio-economic impacts of COVID-19. As this is a cross-sectional survey, we are unable to infer causality in exploring associations of pandemic impact on household financial security and maternal psychosocial wellbeing. It is possible that mothers with pre-existing mental health problems were more likely to be affected financially by the pandemic. Longitudinal research mentioned previously however suggests that the financial impacts of the pandemic are likely to lead to a worsening of mental health problems.

## Conclusions

We find that the COVID-19 pandemic has increased the vulnerability of UK mothers with infants, particularly those in pre-existing low socio-economic positions, including lone and younger mothers, through rising financial insecurity and associated poorer psychosocial wellbeing. Increased requirements of having children at home with childcare and school closures put further strain on households, particularly for working mothers. Although several financial support packages were rolled out during the pandemic, similarly to others, our findings indicate that these may have been insufficient for many families, especially the poorest. This adds to other burdens of stress reported by new mothers during the period and may have important implications for their children's development and health. Policies going forward need to consider ways to better support families, particularly those with younger children and from lower income groups, to avoid increasing socio-economic inequalities and the burden of poor mental health further.

## Supporting information

**S1 Table. Principal Component Analysis (PCA) Maternal recent psychosocial wellbeing component characteristics (N = 1134).**
(PDF)

**S2 Table. Impact of COVID-19 on household ability to cover expenses by maternal socio-demographic characteristics in the COVID-19 New Mum Study; % (n).**
(PDF)

**S3 Table. Impact of the pandemic on partner employment by household income in the COVID-19 New Mum Study; % (n).**
(PDF)

**S4 Table. Reported impact of the pandemic on employment and related factors in the UK COVID-19 New Mum Study.**
(PDF)

## Author Contributions

**Conceptualization:** Emeline Rougeaux, Sarah Dib, Adriana Vázquez-Vázquez, Mary S. Fewtrell, Jonathan C. K. Wells.

**Formal analysis:** Emeline Rougeaux.

**Visualization:** Emeline Rougeaux.

**Writing – original draft:** Emeline Rougeaux.

**Writing – review & editing:** Emeline Rougeaux, Sarah Dib, Adriana Vázquez-Vázquez, Mary S. Fewtrell, Jonathan C. K. Wells.

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
