## [Decision Letter · Decision Letter 0]

7 Apr 2022

PGPH-D-22-00025

Socio-economic impacts of the COVID-19 pandemic on new mothers and associations with psychosocial wellbeing: findings from the UK COVID-19 New Mum Study.

Dear Dr. Rougeaux,

Thank you for submitting your manuscript to PLOS Global Public Health. After careful consideration, we feel that it has merit but does not fully meet PLOS Global Public Health’s publication criteria as it currently stands. Therefore, we invite you to submit a revised version of the manuscript that addresses the points raised during the review process, which you will find below.

We look forward to receiving your revised manuscript.

Kind regards,

Karen D. Cowgill, PhD, MSc

Academic Editor

Journal Requirements:

1. Your co-authors:

Sarah Dib -sarah.dib.15@ucl.ac.uk

Mary S. Fewtrell -m.fewtrell@ucl.ac.uk

Jonathan C.K. Wells -jonathan.wells@ucl.ac.uk

,have not confirmed authorship of the manuscript. We have resent them the authorship confirmation email; however please check that the above email address for them is correct and follow up personally to ensure they confirm. 

Please note that we cannot proceed your manuscript  until we have received confirmations from all co-authors.

2. In the online submission form you indicate that your data is not available for proprietary reasons and have provided a contact point for accessing this data. Please note that your current contact point is a co-author on this manuscript. According to our Data Policy, the contact point must not be an author on the manuscript and must be a third party. Please revise your data statement to a non-author institutional point of contact, such as a data access or ethics committee, and send this to us via return email. Please also include contact information for the third party organization, and please include the full citation of where the data can be found.

3. Please update the completed 'Competing Interests' statement, including any COIs declared by your co-authors. If you have no competing interests to declare, please state "The authors have declared that no competing interests exist". Otherwise please declare all competing interests beginning with the statement "I have read the journal's policy and the authors of this manuscript have the following competing interests:"

Additional Editor Comments (if provided):

Title: specify period, study type

Abstract: how were data collected? over what period? from text, this appears to be a convenience sample. Include N here. Give exact p-values. define "redundancy" as this term may not be familiar to international readers. [[Clarify whether the concern is for new mothers or their infants both or primarily for the long-term health and development of infants? ]]

Line 49: update as of early 2022

Line 188: all these characteristics as well as other study responses are self-reported and not confirmed, correct?

Table titles: add UK to all, year

Table 1: add a column with the frequency of these characteristics in the UK population overall (at line 387 you acknowledge that the sample did not reflect UK population). Please also consider possible limitations of language access as well as internet/social media access and use.

Lines 392-394: here you mention that there were missing values for many variables -- these are not reflected in the tables showing the analyses; please specify the N for each analysis -- this info is in supporting info but not in the main article.

Consider reporting some of the qualitative statements from table S4 in the main article.

As noted above, this study is based on a convenience sample -- please temper conclusions accordingly.

Reviewers' comments:

Reviewer's Responses to Questions

**Comments to the Author**

1. Does this manuscript meet PLOS Global Public Health’s publication criteria? Is the manuscript technically sound, and do the data support the conclusions? The manuscript must describe methodologically and ethically rigorous research with conclusions that are appropriately drawn based on the data presented.

Reviewer #1: Yes

2. Has the statistical analysis been performed appropriately and rigorously?

Reviewer #1: Yes

3. Have the authors made all data underlying the findings in their manuscript fully available (please refer to the Data Availability Statement at the start of the manuscript PDF file)?

Reviewer #1: Yes

4. Is the manuscript presented in an intelligible fashion and written in standard English?

Reviewer #1: No

5. Review Comments to the Author

Reviewer #1: The authors have masterfully put together an impressive array of analyses on the socio-economic impacts of the COVID-19 pandemic on a critical group of the vulnerable population. These contributions are highly valuable. My overall concern is that the analysis process is not clear since its description is scattered with missing details of specifics of the models fit making it difficult to follow as is in the manuscript. Below are my comments and suggestions

1. Line 28 indicate that it is each of the expenses listed and not either so that it is easy to understand the results as presented on line 36

2. Line 30 it is not clear what is meant by socio-economic position at this point. Include a definition. I would suggest that you mention the types of models fit for each association and the rationale rather than just including the measures eg we explored the associations of i. and socio-economic position using a logistic regression model to obtain odds ratios etc

3. Line 43 whose health and development do you refer to here I would presume child? Also, the conclusion is too general. There are support packages already in place so what needs to be done is being more strategic to allocate these resources to more vulnerable populations in this case new mothers. This needs to be reflected in the conclusion provided in the abstract as it is in the last section of the paper.

4. Lines 47-49 in the introduction need to be referenced, what is the source of the information used?

5. I suggest that you provide Lines 120-140 as a table with a column of the indicator and the operational definition as used in the study which would be easier to follow. Include the psychosocial wellbeing on the table and then retain information on how it was defined as is in the manuscript.

6. Only describe the PCA process used to construct maternal psychosocial wellbeing and move the PCA results presented in lines 148-159 after S1 table. As is, it distracts the flow of manuscript by reporting results in methods section.

7. The analyses section is a bit confusing. This section could be written better to have an easy flow of the steps taken without subdividing into sections and then repeating tests that were done across all the associations explored. Also, at this point it is not clear what models were fit for each association and the rationale behind it. It is critical to ensure that this section is described clearly for replication. Define what confounders you used. Kindly rewrite this to reflect the stepwise approach used during the analysis.

8. Line 184 state the STATA license used

9. With regards to the sample characteristics section, you had stated on the methods section that disparities of socio-economic groups and stresses were assessed using proportions & chi-square test. This has not been reported here

10. Review what happened to the border lines of table 4

11. On Line 337 kindly clarify which of your findings suggest that the long-term economic effects of the pandemic are likely to increase the impact on families further.

12. Line 389, do you mean lack of representativeness because power might allude to sample size which I don't think is the case here

13. Well written conclusion that links well to the findings. Align the references well, at the moment they appear disorderly

14. I might be colour blind but Figure 2 the blue colours are difficult to distinguish

6. PLOS authors have the option to publish the peer review history of their article (what does this mean?). If published, this will include your full peer review and any attached files.

**Do you want your identity to be public for this peer review?** For information about this choice, including consent withdrawal, please see our Privacy Policy.

Reviewer #1: No

---

## [Decision Letter · Decision Letter 1]

31 May 2022

Socio-economic impacts of the COVID-19 pandemic on new mothers and associations with psychosocial wellbeing: findings from the UK COVID-19 New Mum Online Observational Study (May 2020-June 2021).

PGPH-D-22-00025R1

Dear Ms Rougeaux,

We are pleased to inform you that your manuscript 'Socio-economic impacts of the COVID-19 pandemic on new mothers and associations with psychosocial wellbeing: findings from the UK COVID-19 New Mum Online Observational Study (May 2020-June 2021).' has been provisionally accepted for publication in PLOS Global Public Health.

Best regards,

Karen D. Cowgill, PhD, MSc

Academic Editor

Reviewer Comments (if any, and for reference):

Reviewer's Responses to Questions

**Comments to the Author**

1. If the authors have adequately addressed your comments raised in a previous round of review and you feel that this manuscript is now acceptable for publication, you may indicate that here to bypass the “Comments to the Author” section, enter your conflict of interest statement in the “Confidential to Editor” section, and submit your "Accept" recommendation.

Reviewer #1: All comments have been addressed

2. Does this manuscript meet PLOS Global Public Health’s publication criteria? Is the manuscript technically sound, and do the data support the conclusions? The manuscript must describe methodologically and ethically rigorous research with conclusions that are appropriately drawn based on the data presented.

Reviewer #1: Yes

3. Has the statistical analysis been performed appropriately and rigorously?

Reviewer #1: Yes

4. Have the authors made all data underlying the findings in their manuscript fully available (please refer to the Data Availability Statement at the start of the manuscript PDF file)?

Reviewer #1: Yes

5. Is the manuscript presented in an intelligible fashion and written in standard English?

Reviewer #1: Yes

6. Review Comments to the Author

Reviewer #1: The manuscript reads better now and all the comments have been addressed adequately

7. PLOS authors have the option to publish the peer review history of their article (what does this mean?). If published, this will include your full peer review and any attached files.

**Do you want your identity to be public for this peer review?** For information about this choice, including consent withdrawal, please see our Privacy Policy.

Reviewer #1: No
